# IsoNN: Isomorphic Neural Network for Graph Representation Learning and Classification

## Abstract

Deep learning models have achieved huge success in numerous fields, such as computer vision and natural language processing. However, unlike such fields, it is hard to apply traditional deep learning models on the graph data due to the 'node-orderless' property. Normally, adjacency matrices will cast an artificial and random node-order on the graphs, which renders the performance of deep models on graph classification tasks extremely erratic, and the representations learned by such models lack clear interpretability. To eliminate the unnecessary node-order constraint, we propose a novel model named **Iso**morphic **N**eural **N**etwork (IsoNN), which learns the graph representation by extracting its isomorphic features via the graph matching between input graph and templates. IsoNN has two main components: graph isomorphic feature extraction component and classification component. The graph isomorphic feature extraction component utilizes a set of subgraph templates as the kernel variables to learn the possible subgraph patterns existing in the input graph and then computes the isomorphic features. A set of permutation matrices is used in the component to break the node-order brought by the matrix representation. Three fully-connected layers are used as the classification component in IsoNN. Extensive experiments are conducted on benchmark datasets, the experimental results can demonstrate the effectiveness of IsoNN, especially compared with both classic and state-of-the-art graph classification methods.

## 1    Introduction

The graph structure is attracting increasing interests because of its great representation power on various types of data. Researchers have done many analyses based on different types of graphs, such as social networks, brain networks and biological networks. In this paper, we will focus on the binary graph classification problem, which has extensive applications in the real world. For example, one may wish to identify the social community categories according to the users' social interactions (Gao et al., 2017), distinguish the brain states of patients via their brain networks (Wang et al., 2017), and classify the functions of proteins in a biological interaction network (Hamilton et al., 2017).

To address the graph classification task, many approaches have been proposed. One way to estimate the usefulness of subgraph features is feature evaluation criteria based on both labeled and unlabeled graphs (Kong & Yu, 2010). Some other works also proposed to design a pattern exploration approach based on pattern co-occurrence and build the classification model (Jin et al., 2009) or develop a boosting algorithm (Wu et al., 2014). However, such works based on BFS or DFS cannot avoid computing a large quantity of possible subgraphs, which causes high computational complexity though the explicit subgraphs are maintained. Recently, deep learning models are also widely used to solve the graph-oriented problems. Although some deep models like MPNN (Gilmer et al., 2017) and GCN (Kipf & Welling, 2016) learn implicit structural features, the explicit structural information cannot be maintained for further research. Besides, most existing works on graph classification use the aggregation of the node features in graphs as the graph representation (Xu et al., 2018; Hamilton et al., 2017), but simply doing aggregation on the whole graph cannot capture the substructure precisely. While there are other models can capture the subgraphs, they often need more complex computation and mechanism (Wang et al., 2017; Narayanan et al., 2017) or need additonal node labels to find the subgraph strcuture (Gaüzere et al., 2012; Shervashidze et al., 2011).

However, we should notice that when we deal with the graph-structured data, different node-orders will result in very different adjacency matrix representations for most existing deep models which take the adjacency matrices as input if there is no other information on graph. Therefore, compared with the original graph, matrix naturally poses a redundant constraint on the graph node-order. Such a node-order is usually unnecessary and manually defined. The different graph matrix representations brought by the node-order differences may render the learning performance of the existing models to be extremely erratic and not robust. Formally, we summarize the encountered challenges in the graph classification problem as follows:

- **Explicit useful subgraph extraction.** The existing works have proposed many discriminative models to discover useful subgraphs for graph classification, and most of them require manual efforts. Nevertheless, how to select the contributing subgraphs automatically without any additional manual involvement is a challenging problem.

- **Graph representation learning.** Representing graphs in the vector space is an important task since it facilitates the storage, parallelism and the usage of machine learning models for the graph data. Extensive works have been done on node representations (Grover & Leskovec, 2016; Lin et al., 2015; Lai et al., 2017; Hamilton et al., 2017), whereas learning the representation of the whole graph with clear interpretability is still an open problem requiring more explorations.

- **Node-order elimination for subgraphs.** Nodes in graphs are orderless, whereas the matrix representations of graphs cast an unnecessary order on nodes, which also renders the features extracted with the existing learning models, e.g., CNN, to be useless for the graphs. For subgraphs, this problem also exists. Thus, how to break such a node-order constraint for subgraphs is challenging.

- **Efficient matching for large subgraphs.** To break the node-order, we will try all possible node permutations to find the best permutation for a subgraph. Clearly, trying all possible permutaions is a combinatorial explosion problem, which is extremly time-comsuming for finding large subgraph templates. The problem shows that how to accelerate the proposed model for large subgraphs also needs to be solved.

In this paper, we propose a novel model, namely **Iso**morphic **N**eural **N**etwork (ISONN) and its variants, to address the aforementioned challenges in the graph representation learning and classification problem. ISONN is composed of two components: the graph isomorphic feature extraction component and the classification component, aiming at learning isomorphic features and classifying graph instances, respectively. In the graph isomorphic feature extraction component, ISONN automatically learns a group of subgraph templates of useful patterns from the input graph. ISONN makes use of a set of permutation matrices, which act as the node isomorphism mappings between the templates and the input graph. With the potential isomorphic features learned by all the permutation matrices and the templates, ISONN adopts one min-pooling layer to find the best node permutation for each template and one softmax layer to normalize and fuse all subgraph features learned by different kernels, respectively. Such features learned by different kernels will be fused together and fed as the input for the classification component. ISONN further adopts three fully-connected layers as the classification component to project the graph instances to their labels. Moreover, to accelerate the proposed model when dealing with large subgraphs, we also propose two variants of ISONN to gurantee the efficiency.

## 2 RELATED WORK

Our work relates to subgraph mining, graph neural networks, network embedding as well as graph classification. We will discuss them briefly in the followings.

**Subgraph Mining and Graph Kernel Methods:** Mining subgraph features from graph data has been studied for many years. The aim is to extract useful subgraph features from a set of graphs by adopting some specific criteria. One classic unsupervised method (i.e., without label information) is gSpan (Yan & Han, 2002), which builds a lexicographic order among graphs and map each graph to a unique minimum DFS code as its canonical label. GRAMI (Elseidy et al., 2014) only stores templates of frequent subgraphs and treat the frequency evaluation as a constraint satisfaction problem to find the minimal set. For the supervised model (i.e., with label information), CORK utilizes labels to guide the feature selection, where the features are generated by gSpan (Thoma et al., 2009). Due to the mature development of the sub-graph mining field, subgraph mining methods have also

been adopted in life sciences (Mrzic et al., 2018). Moreover, several parallel computing based methods (Qiao et al., 2018; Hill et al., 2012; Lin et al., 2014) have proposed to reduce the time cost. On the other hand, graph kernel methods are also applied to discover the subgraph structures (Kashima et al., 2003; Vishwanathan et al., 2010; Gaüzere et al., 2012; Shervashidze et al., 2011). Among them, most existing works focus on the graph with node labels and the kernels methods only computes the similarity between pairwise graphs. Yet, in this paper, we are handling the graph without node labels. Moreover, we can not only compute the similarity between pairwise graphs but also learn subgraph templates, which can be further analyzed.

**Graph Neural Network and Network Embedding:** Graph Neural Networks (Monti et al., 2017; Atwood & Towsley, 2016; Masci et al., 2015; Kipf & Welling, 2016; Battaglia et al., 2018) have been studied in recent years because of the prosperity of deep learning. Traditional deep models cannot be directly applied to graphs due to the special data structure. The general graph neural model MoNet (Monti et al., 2017) employs CNN architectures on non-Euclidean domains such as graphs and manifold. The GCN proposed in (Kipf & Welling, 2016) utilizes the normalized adjacency matrix to learn the node features for node classification; (Bai et al., 2018) proposes the multi-scale convolutional model for pairwise graph similarity with a set matching based graph similarity computation. However, these existing works based on graph neural networks all fail to investigate the node-orderless property of the graph data and to maintain the explicit structural information. Another important topic related to this paper is network embedding (Bordes et al., 2013; Lin et al., 2015; Lai et al., 2017; Abu-El-Haija et al., 2018; Hamilton et al., 2017), which aims at learning the feature representation of each individual node in a network based on either the network structure or attribute information. Distinct from these network embedding works, the graph representation learning problem studied in this paper treats each graph as an individual instance and focuses on learning the representation of the whole graph instead.

**Graph Classification:** Graph classification is an important problem with many practical applications. Data like social networks, chemical compounds, brain networks can be represented as graphs naturally and they can have applications such as community detection (Zhang et al., 2018), anti-cancer activity identification (Kong et al., 2013; Kong & Yu, 2010) and Alzheimer's patients diagnosis (Tong et al., 2017; 2015) respectively. Traditionally, researchers mine the subgraphs by DFS or BFS (Saigo et al., 2009; Kong et al., 2013), and use them as the features. With the rapid development of deep learning (DL), many works are done based on DL methods. GAM builds the model by RNN with self-attention mechanism (Lee et al., 2018). DCNN extend CNN to general graph-structured data by introducing a 'diffusion-convolution' operation (Atwood & Towsley, 2016).

## 3 TERMINOLOGY AND PROBLEM DEFINITION

In this section, we will define the notations and the terminologies used in this paper and give the formulation for the graph classification problem.

### 3.1 NOTATIONS

In the following sections, we will use lower case letters like $x$ to denote scalars, lower case bold letters (e.g. $\mathbf{x}$) to represent vectors, bold-face capital letters (e.g. $\mathbf{X}$) to show the matrices. For tensors or sets, capital calligraphic letters are used to denote them. We use $\mathbf{x}_i$ to represent the $i$-th element in $\mathbf{x}$. Given a matrix $\mathbf{X}$, we use $\mathbf{X}(i, j)$ to express the element in $i$-th row and $j$-th column. For $i$-th row vector and $j$-th column vector, we use $\mathbf{X}(i, :)$ and $\mathbf{X}(:, j)$ to denote respectively. Moreover, notations $\mathbf{x}^\top$ and $\mathbf{X}^\top$ denote the transpose of vector $\mathbf{x}$ and matrix $\mathbf{X}$ respectively. Besides, the $F$-norm of matrix $\mathbf{X}$ can be represented as $\|\mathbf{X}\|_F = (\sum_{i,j} |X_{i,j}|^2)^{\frac{1}{2}}$.

### 3.2 PROBLEM FORMULATION

Many real-world inter-connected data can be formally represented as the graph-structured data.

**DEFINITION 1** *(Graph): Formally, a graph can be represented as $G = (\mathcal{V}, \mathcal{E})$, where the sets $\mathcal{V}$ and $\mathcal{E}$ denote the nodes and links involved in the graph, respectively.*
Some representative examples include the human brain graphs (where the nodes denote brain regions and links represent the correlations among these regions), biological molecule graphs (with the nodes represent the atoms and links denote the atomic bonds), as well as the geographical graphs in the offline world (where the nodes denote the communities and the links represent the commute

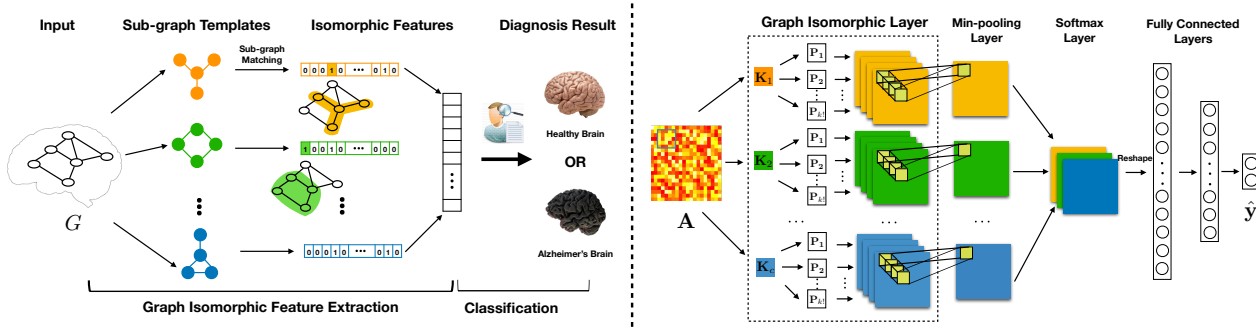

Figure 1: IsoNN Framework Architecture. (The left subplot provides the outline of the proposed framework, including the *graph isomorphic feature extraction* component and the *classification* component respectively. Meanwhile, the right subplot illustrates the detailed architecture of the proposed framework, where the *graph isomorphic features* are extracted with the *graph isomorphic layer*, *min-pooling layer* and *softmax layer*, and the graphs are further classified with three fully-connected layers.)

routes among communities). Meanwhile, many concrete real-world application problems, e.g., brain graph based patient disease diagnosis, molecule function classification and community vibrancy prediction can also be formulated as the graph classification problems.

**Problem Definition**: Formally, given a graph set $\mathcal{G} = \{G_1, G_2, \cdots, G_n\}$ with a small number of labeled graph instances, the graph classification problem aims at learning a mapping, i.e., $f : \mathcal{G} \rightarrow \mathcal{Y}$, to project each graph instance into a pre-defined label space $\mathcal{Y} = \{+1, -1\}$.

In this paper, we will take the graph binary classification as an example to illustrate the problem setting for ISONN. A simple extension of the model can be applied to handle more complicated learning scenarios with multi-class or multi-label as well.

## 4 PROPOSED METHOD

The overall architecture of ISONN is shown in Figure 1. The ISONN framework includes two main components: graph isomorphic feature extraction component and classification component. The graph isomorphic feature extraction component includes a graph isomorphic layer, a min-pooling layer as well as a softmax layer and the classification component is composed by three fully-connected layers. They will be discussed in detail in the following subsections.

### 4.1 GRAPH ISOMORPHIC FEATURE EXTRACTION COMPONENT

Graph isomorphic feature extraction component targets at learning the graph features. To achieve that objective, ISONN adopts an automatic feature extraction strategy for graph representation learning. In ISONN, one graph isomorphic feature extraction component involves three layers: the graph isomorphic layer, the min-pooling layer and the softmax layer. In addition, we can further construct a deep graph isomorphic neural network by applying multiple isomorphic feature extraction components on top of each other, *i.e.*, apply the combination of "graph isomorphic layer, min pooling layer, softmax layer" several times. For the second and latter components, they will be used on every feature matrix learned by the combination of channels of all former components.

### 4.1.1 GRAPH ISOMORPHIC LAYER

Graph isomorphic layer is the first effective layer in deep learning that handles the node-order restriction in graph representations. Assume we have a graph $G = \{\mathcal{V}, \mathcal{E}\}$, and its adjacency matrix to be $\mathbf{A} \in \mathbb{R}^{|\mathcal{V}| \times |\mathcal{V}|}$. In order to find the existence of specific subgraph patterns in the input graph, ISONN matches the input graph with a set of subgraph templates. Each template is denoted as a kernel variable $\mathbf{K}_i \in \mathbb{R}^{k \times k}, \forall i \in \{1, 2, \cdots, c\}$. Here, $k$ denotes the node number in subgraphs and $c$ is the channel number (i.e., total template count). Meanwhile, to match one template with regions in the input graph (i.e., sub-matrices in $\mathbf{A}$), we use a set of permutation matrices, which map both rows and columns of the kernel variable to the subgraphs effectively. The permutation matrix can be represented as $\mathbf{P} \in \{0, 1\}^{k \times k}$ that shares the same dimension with the kernel variable. Therefore, given a kernel $\mathbf{K}_i$ and a sub-matrix $\mathbf{M}_{(s,t)} \in \mathbb{R}^{k \times k}$ in $\mathbf{A}$ (i.e., a region in the input graph $G$ and $s, t \in \{1, 2, \cdots, (|\mathcal{V}| - k + 1)\}$ denotes a starting index pair in $\mathbf{A}$), there may exist $k!$ different such permutation matrices. The optimal should be the matrix $\mathbf{P}^*$ that minimizes the following term.

$$\mathbf{P}^* = \arg\min_{\mathbf{P} \in \mathcal{P}} \left\| \mathbf{P} \mathbf{K}_i \mathbf{P}^\top - \mathbf{M}_{(s,t)} \right\|_F^2 , \tag{1}$$

where $\mathcal{P} = \{\mathbf{P}_1, \mathbf{P}_2, \cdots, \mathbf{P}_{k!}\}$ covers all the potential permutation matrices. Formally, the isomorphic feature extracted based on the kernel $\mathbf{K}_i$ for the regional sub-matrix $\mathbf{M}_{(s,t)}$ in $\mathbf{A}$ can be represented as

$$z_{i,(s,t)} = \left\|\mathbf{P}^*\mathbf{K}_i(\mathbf{P}^*)^\top - \mathbf{M}_{(s,t)}\right\|_F^2 = \min\{\left\|\mathbf{P}\mathbf{K}_i\mathbf{P}^\top - \mathbf{M}_{(s,t)}\right\|_F^2\}_{\mathbf{P}\in\mathcal{P}}$$
$$= \min(\bar{\mathbf{z}}_{i,(s,t)}(1:k!)), \tag{2}$$

where vector $\bar{\mathbf{z}}_{i,(s,t)} \in \mathbb{R}^{k!}$ contains entry $\bar{\mathbf{z}}_{i,(s,t)}(j) = \left\|\mathbf{P}_j\mathbf{K}_i\mathbf{P}_j^\top - \mathbf{M}_{(s,t)}\right\|_F^2, \forall j \in \{1, 2, \cdots, k!\}$ denoting the isomorphic features computed by the $j$-th permutation matrix $\mathbf{P}_j \in \mathcal{P}$.

As indicated by the Figure 1, IsoNN computes the final isomorphic features for the kernel variable $\mathbf{K}_i$ via two steps: (1) computing all the potential isomorphic features via different permutation matrices with the graph isomorphic layer, and (2) identifying and fusing the optimal features with the min-pooling layer and softmax layer to be introduced as follows. By shifting one kernel matrix $\mathbf{K}_i$ on regional sub-matrices, IsoNN extracts the isomorphic features on the matrix $\mathbf{A}$, which can be denoted as a 3-way tensor $\bar{\mathcal{Z}}_i \in \mathbb{R}^{k!\times(|\mathcal{V}|-k+1)\times(|\mathcal{V}|-k+1)}$, where $\bar{\mathcal{Z}}_i(1:k!,s,t) = \bar{\mathbf{z}}_{i,(s,t)}(1:k!)$. In a similar way, we can also compute the isomorphic feature tensors based on the other kernels, which can be denoted as $\bar{\mathcal{Z}}_1, \bar{\mathcal{Z}}_2, \cdots, \bar{\mathcal{Z}}_c$ respectively.

### 4.1.2 MIN-POOLING LAYER

Given the tensor $\bar{\mathcal{Z}}_i$ computed by $\mathbf{K}_i$ in the graph isomorphic layer, IsoNN will identify the optimal permutation matrices via the min-pooling layer. Formally, we can represent results of the optimal permutation selection with $\bar{\mathcal{Z}}_i$ as matrix $\mathbf{Z}_i$:

$$\mathbf{Z}_i(s,t) = \min\{\bar{\mathcal{Z}}_i(1:k!,s,t)\}. \tag{3}$$

The min-pooling layer learns the optimal matrix $\mathbf{Z}_i$ for kernel $\mathbf{K}_i$ along the first dimension (i.e., the dimension indexed by different permutation matrices), which can effectively identify the isomorphic features created by the optimal permutation matrices. For the remaining kernel matrices, we can also achieve their corresponding graph isomorphic feature matrices as $\mathbf{Z}_1, \mathbf{Z}_2, \cdots, \mathbf{Z}_c$ respectively.

### 4.1.3 SOFTMAX LAYER

Based on the above descriptions, a perfect matching between the subgraph templates with the input graph will lead to a very small isomorphic feature, e.g., a value approaching to 0. If we feed the small features into the classification component, the useful information will vanish and the relative useless information (*i.e.*, features learned by the subgraphs dismatch the kernels) dominates the learning feature vector in the end. Meanwhile, the feature values computed in Equation (3) can also be in different scales for different kernels. To effectively normalize these features, we propose to apply the softmax function to matrices $\mathbf{Z}_1, \mathbf{Z}_2, \cdots, \mathbf{Z}_c$ across all $c$ kernels. Compared with the raw features, e.g., $\mathbf{Z}_i$, softmax as a non-linear mapping can also effectively highlight the useful features in $\mathbf{Z}_i$ by rescaling them to relatively larger values especially compared with the useless ones. Formally, we can represent the fused graph isomorphic features after rescaling by all the kernels as a 3-way tensor $\mathcal{Q}$, where slices along first dimension can be denoted as:

$$\mathcal{Q}(i,:,:) = \hat{\mathbf{Z}}_i \text{, where } \hat{\mathbf{Z}}_i = softmax(-\mathbf{Z}_i), \ \forall i \in \{1,\ldots,c\}. \tag{4}$$

## 4.2 CLASSIFICATION COMPONENT

After the isomorphic feature tensor $\mathcal{Q}$ is obtained, we feed it into a classification component. Let $\mathbf{q}$ denote the flattened vector representation of feature tensor $\mathcal{Q}$, and we pass it to three fully-connected layers to get the predicted label vector $\hat{\mathbf{y}}$. For the graph binary classification, suppose we have the ground truth $\mathbf{y} = (y_1^g, y_2^g)$ and the predicted label vector $\hat{\mathbf{y}}^g = (\hat{y}_1^g, \hat{y}_2^g)$ for the sample $g$ from the training batch set $\mathcal{B}$. We use cross-entropy as the loss function in IsoNN. Formally, the fully-connected (FC) layers and the objective function can be represented as follows respectively:

$$\text{FC Layers:} \begin{cases} \mathbf{d}_1 &= \sigma(\mathbf{W}_1\mathbf{q} + \mathbf{b}_1), \\ \mathbf{d}_2 &= \sigma(\mathbf{W}_2\mathbf{d}_1 + \mathbf{b}_2), \\ \hat{\mathbf{y}} &= \sigma(\mathbf{W}_3\mathbf{d}_2 + \mathbf{b}_3), \end{cases} \text{Objective Function: } \mathcal{L} = -\sum_{g\in\mathcal{B}}\sum_{j=1}^{2} y_j^g \log \hat{y}_j^g, \tag{5}$$

where $\mathbf{W}_i$ and $\mathbf{b}_i$ represent the weights and biases in $i$-th layer respectively for $i \in \{1, 2, 3\}$. The $\sigma$ denotes the adopted the relu activation function. To train the proposed model, we adopt the back propagation algorithm to learn both the subgraph templates and the other involved variables.

### 4.3 MORE DISCUSSIONS ON GRAPH ISOMORPHIC FEATURE EXTRACTION IN ISONN

Before introducing the empirical experiments to test the effectiveness of IsoNN, we would like to provide more discussions about the computation time complexity of the graph isomorphic feature extraction component involved in IsoNN. Formally, given the input graph $G$ with $n = |\mathcal{V}|$ nodes, by shifting the kernel variables (of size $k \times k$) among the dimensions of the corresponding graph adjacency matrix, we will be able to obtain $(n-k+1)^2$ regional sub-matrices (or $\mathcal{O}(n^2)$ regional sub-matrices for notation simplicity). Here, we assume IsoNN has only one isomorphic layer involving $c$ different kernels. In the forward propagation, the introduced time cost in computing the graph isomorphic features can be denoted as $\mathcal{O}(ck!k^3n^2)$, where term $k!$ is introduced in enumerating all the potential permutation matrices and $k^3$ corresponds to the matrix multiplication time cost.

According to the notation, we observe that $n$ is fixed for the input graph. Once the kernel channel number $c$ is decided, the time cost notation will be mainly dominated by $k$. To lower down the above time complexity notation, in this part, we propose to further improve IsoNN from two perspectives: (1) compute the optimal permutation matrix in a faster manner, and (2) use deeper model architectures with small-sized kernels.

#### 4.3.1 FAST PERMUTATION MATRIX COMPUTATION

Instead of enumerating all the permutation matrices in the graph isomorphic feature extraction as indicated by Equations (2)-(3), here we introduce a fast way to compute the optimal permutation matrix for the provided kernel variable matrix, e.g., $\mathbf{K}_i$, and input regional sub-matrix, $\mathbf{M}_{(s,t)}$, directly according to the following theorem.

**THEOREM 1** *Formally, let the kernel variable $\mathbf{K}_i$ and the input regional sub-matrix $\mathbf{M}_{(s,t)}$ be $k \times k$ real symmetric matrices with $k$ distinct eigenvalues $\alpha_1 > \alpha_2 > \cdots > \alpha_k$ and $\beta_1 > \beta_2 > \cdots > \beta_k$, respectively, and their eigendecomposition be represented by*

$$\mathbf{K}_i = \mathbf{U}_{K_i}\mathbf{\Lambda}_{K_i}\mathbf{U}_{K_i}^\top, \text{ and } \mathbf{M}_{(s,t)} = \mathbf{U}_{M_{(s,t)}}\mathbf{\Lambda}_{M_{(s,t)}}\mathbf{U}_{M_{(s,t)}}^\top \tag{6}$$

*where $\mathbf{U}_{K_i}$ and $\mathbf{U}_{M_{(s,t)}}$ are orthogonal matrices of eigenvectors and $\mathbf{\Lambda}_{K_i} = diag(\alpha_j), \mathbf{\Lambda}_{M_{(s,t)}} = diag(\beta_j)$. The minimum of $||\mathbf{P}\mathbf{K}_i\mathbf{P}^\top - \mathbf{M}_{(s,t)}||^2$ is attained for the following $\mathbf{P}$'s:*

$$\mathbf{P}^* = \mathbf{U}_{M_{(s,t)}}\mathbf{S}\mathbf{U}_{K_i}^\top \tag{7}$$

*where $\mathbf{S} \in \mathcal{S} = \{diag(s_1, s_2, \cdots, s_k)|s_i = 1 \text{ or } -1\}$.*

The proof of the theorem will be provided in appendix. In computing the optimal permutation matrix $\mathbf{P}^*$, trials of different $\mathbf{S}$ will be needed. Meanwhile, to avoid such time costs, we introduce to take the upper bound value of $\mathbf{U}_{M_{(s,t)}}\mathbf{S}\mathbf{U}_{K_i}^\top$ as the approximated optimal permutation matrix instead, which together with the corresponding optimal feature $z_{i,(s,t)}$ can be denoted as follows:

$$\mathbf{P}^* = |\mathbf{U}_{\mathbf{M}_{(s,t)}}||\mathbf{U}_{K_i}^\top| \text{ and } z_{i,(s,t)} = ||\mathbf{P}^*\mathbf{K}(\mathbf{P}^*)^\top - \mathbf{M}_{(s,t)}||^2, \tag{8}$$

where $|\cdot|$ denotes the absolute value operator and $|\mathbf{U}_{\mathbf{M}_{(s,t)}}||\mathbf{U}_{K_i}^\top| \geq \mathbf{U}_{M_{(s,t)}}\mathbf{S}\mathbf{U}_{K_i}^\top$ hold for $\forall \mathbf{S} \in \mathcal{S}$.

By replacing Equations (2)-(3) with Equation (7), we can compute the optimal graph isomorphic feature for the kernel $\mathbf{K}_i$ and input regional sub-matrix $\mathbf{M}_{(s,t)}$ with a much lower time cost. Furthermore, since the eigendecomposition time complexity of a $k \times k$ matrix is $\mathcal{O}(k^3)$, based on the above theorem, we will be able to lower down the total time cost in graph isomorphic feature extraction to $\mathcal{O}(ck^3n^2)$, which can be optimized with the method introduced in the following subsection.

#### 4.3.2 DEEP GRAPH ISOMORPHIC FEATURE EXTRACTION

Since graph isomorphic layer is the main functional layer, we simply use multi-layer for short to denote the mutiple graph isomorphic feature extraction components (*i.e.*, deep model). We also provide an example of a deep model in appendix. Here, we will illustrate the advantages of deep IsoNN model with small-sized kernels compared against shallow IsoNN model with large kernels. In Figure 2, we provide an example two IsoNN models with different model architectures

- the left model has one single layer and 6 kernels, where the kernel size $k = 12$;
- the right model has two layers: layer 1 involves 2 kernels of size 3, and layer 2 involves 3 kernels of size 4.

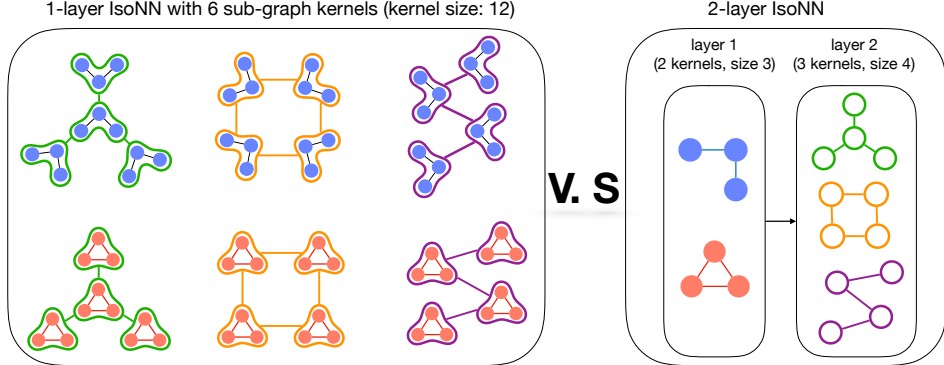

Figure 2: An Illustration of Deep Architecture of ISONN.

By comparing these two different models, we observe that they have identical representation learning capacity. However, the time cost in feature extraction introduced by the left model is much higher than that introduced by the right model, which can be denoted as $\mathcal{O}(6(12^3)n^2)$ and $\mathcal{O}(2(3^3)n^2 + 3(4^3)n^2)$, respectively.

Therefore, for the ISONN model, we tend to use small-sized kernels. Formally, according to the fast method provided in the previous part, given a 1-layer ISONN model with $c$ large kernels of size $k$, its graph isomorphic feature extraction time complexity can be denoted as $\mathcal{O}(ck^3n^2)$. Inspired by Figure 2, without affecting the representation capacity, such a model can be replaced by a $\max\{\lceil \log_2^c \rceil, \lceil \log_3^k \rceil\}$-layers deep ISONN model instead, where each layer involves 2 kernels of size 3. The graph isomorphic feature extraction time complexity of the deep model will be $\mathcal{O}\left((\max\{\lceil \log_2^c \rceil, \lceil \log_3^k \rceil\}) \cdot 2 \cdot 3^3 n^2\right)$ (or $\mathcal{O}\left((\max\{\lceil \log^c \rceil, \lceil \log^k \rceil\}) \cdot n^2\right)$ for simplicity).

## 5 EXPERIMENTS

To evaluate the performance of ISONN, in this section, we will talk about the experimental settings as well as five benchmark datasets. Finally, we will discuss the experimental results with parameter analyses on kernel size , channel number and time complexity.

### 5.1 EXPERIMENTAL SETTINGS

In this subsection, we will use five real-world benchmark datasets: HIV-fMRI (Cao et al., 2015a), HIV-DTI (Cao et al., 2015a), BP-fMRI (Cao et al., 2015b), MUTAG[1] and PTC[1]. Both HIV-fMRI and HIV-DTI have 56 positive instances and 21 negative instances. Also, graph instances in both of them are represented as $90 \times 90$ matrices (Cao et al., 2015a). BP-fMRI has 52 positive and 45 negative instances and each instance is presented by an $82 \times 82$ matrix Cao et al. (2015b). MUTAG and PTC are two datasets which have been widely used in academia Xu et al. (2018); Shervashidze et al. (2011). MUTAG has 125 positive and 63 negative graph instances with graph size $28 \times 28$. PTC is a relative large dataset, which has 152 positive and 192 negative graph instances with graph size $109 \times 109$. With these datasets, we first introduce the comparison methods used in this paper and then talk about the experimental setups and the adopted evaluation metrics in detail.

#### 5.1.1 COMPARISON METHODS

- **ISONN & ISONN-fast** : The proposed method ISONN uses a set of template variables as well as the permutation matrices to extract the isomorphic features and feed these features to the classification component. The variant model named ISONN-fast uses the Equation 8 to compute the optimal permutation matrices and other settings remain unchanged.

- **Freq**: The method uses the top-$k$ frequent subgraphs as its features. This is also an unsupervised feature selection method based on frequency.

- **AE**: We use the autoencoder model (AE) (Vincent et al., 2010) to get the features of graphs without label information. It is an unsupervised learning method, which learns the latent representations of connections in the graphs without considering the structural information.

- **CNN:** It is the convolutional model (Krizhevsky et al., 2012) learns the structural information within small regions of the whole graph. We adopt one convolution layer and three fully-connected layer to be the classification module.

---

[1]https://ls11-www.cs.tu-dortmund.de/people/morris/graphkerneldatasets/

Table 1: Classification Results of the Comparison Methods.

| Dataset | Metric | Methods | | | | | | | | |
|---------|--------|------|------|------|------|------|------|------|------------|-------|
| | | Freq | AE | CNN | SDBN | WL | GCN | GIN | IsoNN-fast | IsoNN |
| HIV-fMRI | Acc. | 54.3 | 46.9 | 59.3 | 66.5 | 44.2 | 58.3 | 52.5 | 70.5 | **73.4** |
| | F1 | 58.2 | 35.5 | 66.3 | 66.7 | 27.2 | 56.4 | 35.6 | 69.9 | **72.2** |
| HIV-DTI | Acc. | 64.6 | 62.4 | 54.3 | 65.9 | 47.1 | 57.7 | 55.1 | 60.1 | **67.5** |
| | F1 | 63.9 | 0.0 | 55.7 | 65.6 | 48.4 | 54.4 | 53.6 | 61.9 | **68.3** |
| BP-fMRI | Acc. | 56.8 | 53.6 | 54.6 | 64.8 | 56.2 | 60.7 | 45.4 | 62.3 | **64.9** |
| | F1 | 57.6 | 69.5 | 52.8 | 63.7 | 58.8 | 61.2 | 42.3 | 63.2 | **69.7** |
| MUTAG | Acc. | 76.2 | 50.0 | 81.7 | 54.0 | 52.4 | 63.5 | 54.0 | **83.3** | **83.3** |
| | F1 | 76.9 | 66.7 | 82.3 | 66.7 | 49.9 | 61.9 | 66.7 | **83.6** | 83.0 |
| PTC | Acc. | 57.8 | 50.0 | 54.6 | 50.0 | 49.0 | 49.0 | 49.0 | 53.0 | **59.9** |
| | F1 | 54.9 | **66.5** | 58.9 | **66.5** | 48.9 | 48.9 | 47.5 | 55.8 | 59.9 |

- **SDBN**: A model proposed in (Wang et al., 2017), which reorders the nodes in the graph first and then feeds the reordered graph into an augmented CNN. In this way, it not only learns the structural information but also tries to minimize the effect of the order constraint.

- **WL:** WL (Shervashidze et al., 2011) is a classic algorithm to do the graph isomorphism test. For the graph classification, we computes the similarity scores for test graphs and train graph. The mean of all similarity scores between each test graph and train graphs will be used to do the classification.

- **GCN:** GCN is proposed in (Kipf & Welling, 2016) use the adjacent matrix to learn the implicit structure information in graphs. Here, we use two graph convolutional layers to learn node features and then take the all nodes features as the graph features. One fully-connected layer will be used as graph classification module.

- **GIN:** GIN is proposed in (Xu et al., 2018) can be used to do graph classification with node features. We adopt the same experimental setting as GIN-0 stated in (Xu et al., 2018).

### 5.1.2 EXPERIMENTAL SETUP AND EVALUATION METRICS

In our experiments, to make the results more reliable, we partition the datasets into 3 folds and then set the ratio of train/test according to $2 : 1$, where two folds are treated as the training data and the remaining one is the testing data. We select top-100 features for Freq as stated in (Wang et al., 2017) with a three layer MLP classifier, where the neuron numbers are 1024, 128, 2. For Auto-encoder, we apply the two-layer encoder and two-layer decoder. For the CNN, we apply the one convolutional layer with the size $5 \times 5 \times 50$, a max-pooling layer with kernel size $2 \times 2$, one gating relu layer as activation layer and we set parameters in the classification module the same as Freq classifier. For the SDBN, we set the architecture as follows: we use two layers of "convolution layer + max pooling layer + activation layer " and concatenate a fully connected layer with 100 neurons as well as an activation layer, where the parameters are the same as those in CNN. We also set the dropout rate in SDBN being 0.5 to avoid overfitting. For WL kernel, if the average similarity score for one test graph greater than 0.5, we assign the test graph positive label, otherwise, assign negative label. We follow the setting in (Kipf & Welling, 2016) and (Xu et al., 2018) to do GCN and GIN-0. Here, to make a fair comparison, we will use the adjacency matrices as features (*i.e.*, no node label information) for WL, GCN and GIN. In the experiments, we set the kernel size $k$ in the isomorphic layer for three datasets as 2, 4, 3, 4, 4, respectively, and then set the parameters in classification component the same as those in Freq classifier. In this experiment, we adopt Adam optimizer and the set the learning rate $\eta = 0.001$, and then we report the average results on balanced datasets.

### 5.2 EXPERIMENTAL RESULTS

In this section, we investigate the effectiveness of the learned subgraph-based graph feature representations for graphs. We adopt one isomorphic layer where the kernel size $k = 2$ and channel number $c = 3$ for HIV-fMRI, one isomorphic layer with $(k = 4, c = 2)$, $(k = 3, c = 1)$, $(k = 4, c = 1)$ and $(k = 4, c = 2)$ for the HIV-DTI, BP-fMRI, MUTAG and PTC, respectively. The results are shown in Table 1. From that table, we can observe that IsoNN outperforms all other baseline methods on these all datasets. We need to remark that IsoNN and IsoNN-fast are very close on MUTAG, and IsoNN has the best performance in total on PTC. Compared with Freq, the proposed method achieves a better performance without searching for all possible subgraphs manually. AE has almost the worst performance among all comparison methods. This is because the features learned from

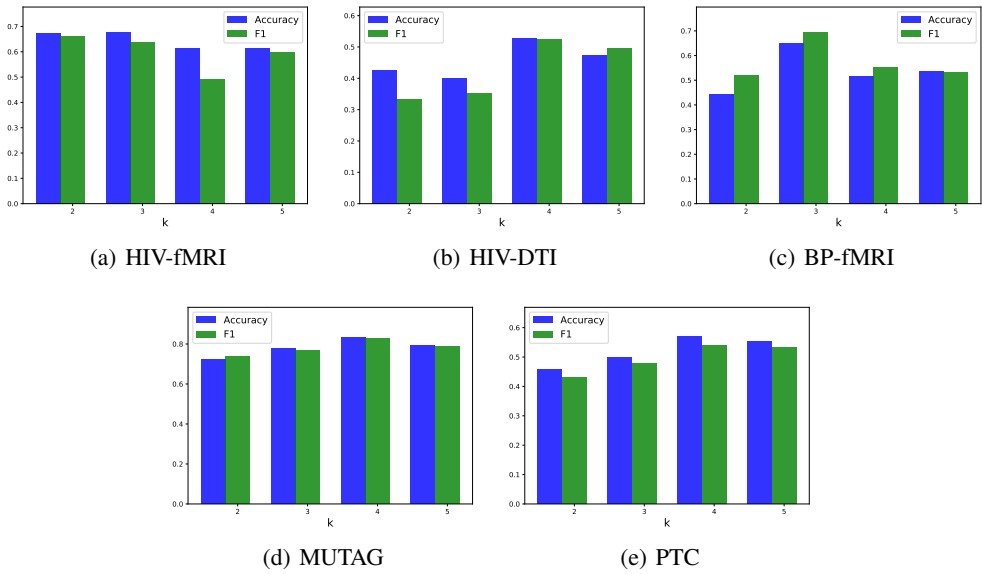

Figure 3: Effectiveness of Different k

AE do not contain any structural information. For HIV-DTI, AE gets 0 in F1. This is because the dataset contains too many zeros, which makes the AE learns trivial features. Also, for PTC, its F1 is higher than other models, but the accuracy only get 50.0, which indicates AE actually have a bad performance since it cannot discriminate the classes of the instances (*i.e.*, predicting all positive classes). CNN performs better than AE but still worse than ISONN. The reason can be that it learns some structural information but fails to learn representative structural patterns. SDBN is designed for brain images, so it may not work for MUTAG and PTC. One possible reason for WL got bad results is the isomorphism test is done on the whole graph, which may lead to erratic results. GCN performs better than GIN but worse than ISONN, showing that GCN can learn some sturctual information without node labels, but GIN cannot work with the adjacency matrix as input. ISONN-fast achieves the best scores on MUTAG and second-best on HIV-fMRI, yet worse than several other methods on other datasets. This can be the approximation on $\mathbf{P}$ may impair the performance. Comparing ISONN with AE, ISONN achieves better results. This means the structural information is more important than only connectivity information for the classification problem. If compared with CNN, the results also show the contribution of breaking the node-order in learning the subgraph templates. Similar to SDBN, ISONN also finds the features from subgraphs, but ISONN gets better performance with more concise architecture. Contrasting with GCN and GIN, ISONN can maintain the explict subgraph structures in graph representations, while the GCN and GIN simply use the aggregation of the neighboring node features, losing the graph-level substructure infomation.

## 5.3 PARAMETER ANALYSIS

To further study the proposed method, we will discuss the effects of different kernel size and channel number in ISONN. The model convergence analysis will be provided in appendix.

- **Kernel Size**: We show the effectiveness of different $k$ in Figure 3. Based on the previous statement, parameter $k$ can affect the final results since it controls the size of learned subgraph templates. To investigate the best kernel size for each dataset, we fix the channel number $c = 1$. As Figure 3 shows, different datasets have different appropriate kernel sizes. The best kernel sizes are 2, 4, 3, 4, 4 for the three datasets, respectively.

- **Channel Number**: We also study the effectiveness of multiple channels (i.e., multiple templates in one layer). To discuss how the channel number influences the results, we choose the best kernel size for each dataset (i.e., 2, 4, 3, 4, 4 respectively). From all subfigures in Figure 4, we can see that the differences among the different channel numbers by using only one isomorphic layer. As shown in Figure 4, ISONN achieves the best results by $c = 3, 2, 1, 1, 2$, respectively, which means the increase of the channel number can improve the performance, but more channels do not necessarily lead to better results. The reason could be the more templates we use, the more complex our model would be. With such a complex model, it is easy to learn an overfitting model on train data, especially when the dataset is quite small. Thus, increasing the channel number can improve the performance but the effectiveness will still depend on the quality and the quantity of the dataset.

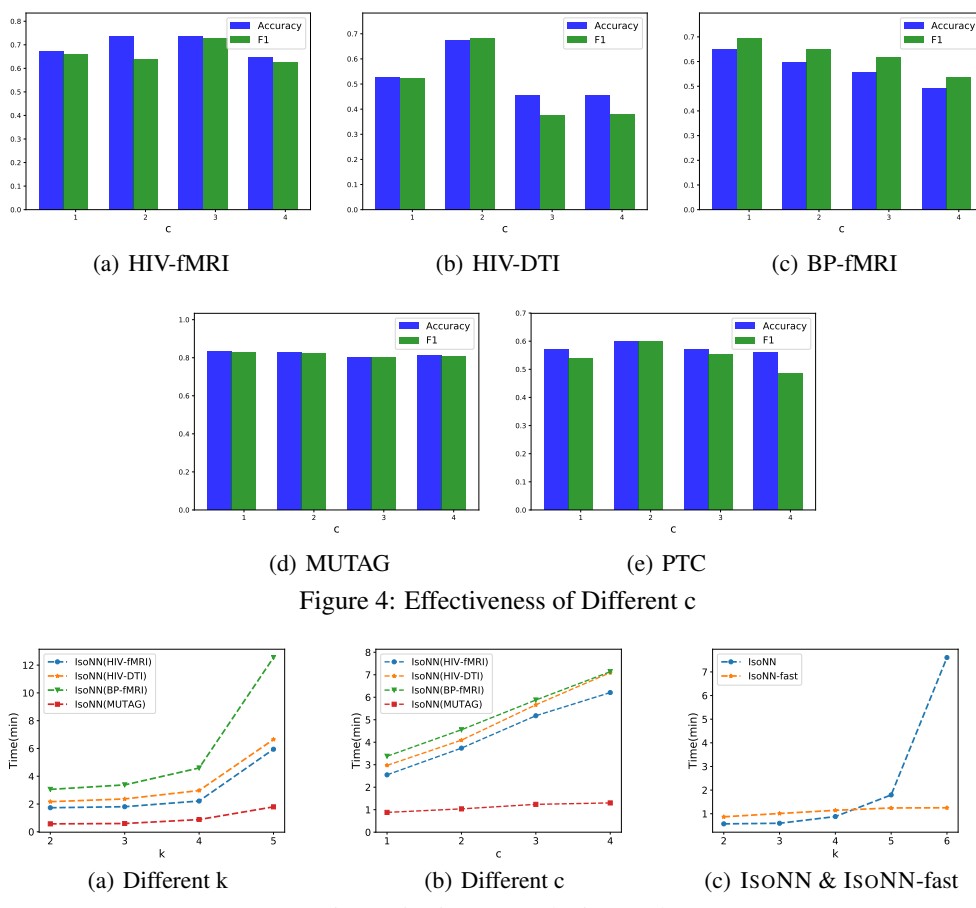

Figure 4: Effectiveness of Different c

Figure 5: Time Complexity Study

## 5.4 TIME COMPLEXITY STUDY

To study the efficiency of ISONN and ISONN-fast, we collect the actual running time on training model, which is shown in Figure 5. In both Figures 5(a) and 5(b) [2], the x-axis denotes its value for $k$ or $c$ and the y-axis denotes the time cost with different parameters. From Figure 5(a), four lines show the same pattern. When the $k$ increases, the time cost grows exponentially. This pattern can be directly explained by the size of the permutation matrix set. When we increase the kernel size by one, the number of corresponding permutation matrices grows exponentially. While changing $c$, shown in Figure 5(b), it is easy to observe that those curves are basically linear with different slopes. This is also natural since whenever we add one channel, we only need to add a constant number of the permutation matrices. To study the efficiency of ISONN-fast, Figure 5(c) shows the running times of ISONN and ISONN-fast on MUTAG. As it shows, ISONN-fast use less time when the kernel size greater than 4, otherwise ISONN and ISONN will have little difference since the eigen decomposition has nearly the same time complexity as calculating all possible node permutaions. The results also verify the theoretical time complexity analysis in 4.3.

## 6 CONCLUSION

In this paper, we proposed a novel graph neural network named ISONN to solve the graph classification problem. ISONN consists of two components: (1) isomorphic component, where a set of permutation matrices is used to break the randomness order posed by matrix representation for a bunch of templates and one min-pooling layer and one softmax layer are used to get the best isomorphic features, and (2) classification component, which contains three fully-connected layers. We further discuss the two efficient variants of ISONN to accelerate the model. Next, we perform the experiments on five real-world datasets. The experimental results show the proposed method outperforms all comparison methods, which demonstrates the superiority of our proposed method. The experimental analysis on time complexity illustrates the efficiency of the ISONN-fast.

---

[2]Since the PTC is a relative large dataset compared with the others, its running time is in different scale compared with the other datasets, which makes the time growth curve of other datasets not obvious. Thus, we don't show the results on PTC.

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

## 7 APPENDIX

### 7.1 PROOF OF THEOREM 1 AND DISCUSSION ABOUT EQUATION (8)

Before giving the proof of Theorem 1, we need to introduce Lemma 1 first.

**LEMMA 1** *If $\mathbf{A}$ and $\mathbf{B}$ are Hermitian matrices with eigenvalues $\alpha_1 \geq \alpha_2 \geq \cdots \geq \alpha_n$ and $\beta_1 \geq \beta_2 \geq \cdots \geq \beta_n$ respectively, then $||\mathbf{A} - \mathbf{B}|| \geq \sum_{i=1}^{n}(\alpha_i - \beta_i)^2$.*

Based on Lemma 1, we can derive the proof of Theorem 1 as follows.

**PROOF 1** *From Lemma 1, Equation 9 holds for any orthogonal matrix $\mathbf{R}$ since the eigenvalues of $\mathbf{R}\mathbf{K}_i\mathbf{R}^\top$ are the same as those of $\mathbf{K}_i$.*

$$||\mathbf{R}\mathbf{K}_i\mathbf{R}^\top - \mathbf{M}_{(s,t)}||^2 \geq \sum_{j=1}^{n}(\alpha_j - \beta_j)^2 \tag{9}$$

*On the other hand, if we use $\mathbf{P}$ in 7, we have*

$$
\begin{aligned}
||\mathbf{P}\mathbf{K}_i\mathbf{P}^\top - \mathbf{M}_{(s,t)}||^2 \quad &= ||\mathbf{U}_{M_{(s,t)}}\mathbf{S}\mathbf{U}_{K_i}^\top\mathbf{U}_{K_i}\mathbf{\Lambda}_{K_i}\mathbf{U}_{K_i}^\top\mathbf{U}_{K_i}\mathbf{S}\mathbf{U}_{M_{(s,t)}}^\top - \mathbf{U}_{M_{(s,t)}}\mathbf{\Lambda}_{M_{(s,t)}}\mathbf{U}_{M_{(s,t)}}^\top||^2 \\
&= ||\mathbf{U}_{M_{(s,t)}}(\mathbf{S}\mathbf{\Lambda}_{K_i}\mathbf{S} - \mathbf{\Lambda}_{M_{(s,t)}})\mathbf{U}_{M_{(s,t)}}^\top||^2 \\
&= ||\mathbf{S}\mathbf{\Lambda}_{K_i}\mathbf{S} - \mathbf{\Lambda}_{M_{(s,t)}}||^2 \\
&= ||\mathbf{\Lambda}_{K_i} - \mathbf{\Lambda}_{M_{(s,t)}}||^2 \\
&= \sum_{j=1}^{n}(\alpha_j - \beta_j)^2
\end{aligned}
\tag{10}
$$

*where we use the equations that $||\mathbf{U}\mathbf{X}|| = ||\mathbf{U}\mathbf{X}^\top|| = ||\mathbf{X}||$ for any orthogonal matrix $\mathbf{U}$ and $\mathbf{S}\mathbf{\Lambda}_{K_i}\mathbf{S} = \mathbf{S}^2\mathbf{\Lambda}_{K_i} = \mathbf{\Lambda}_{K_i}$ since $\mathbf{S}$ and $\mathbf{\Lambda}_{K_i}$ are both orthogonal matrices and $\mathbf{S}^2 = \mathbf{I}$. ∎*

Moreover, it is clear that

$$\mathrm{tr}(\mathbf{P}^\top\mathbf{U}_{M_{(s,t)}}\mathbf{S}\mathbf{U}_{K_i}^\top) \leq \mathrm{tr}\left(\mathbf{P}^\top|\mathbf{U}_{M_{(s,t)}}||\mathbf{U}_{K_i}^\top|\right) \tag{11}$$

because of the elements in $\mathbf{S}$ are either $-1$ or $+1$. Also, since each row vector of $\mathbf{U}_{M_{(s,t)}}$ and $\mathbf{U}_{K_i}$ is unit vector, thus we can have

$$\mathrm{tr}(\mathbf{P}^\top|\mathbf{U}_{M_{(s,t)}}||\mathbf{U}_{K_i}^\top|) \leq n \tag{12}$$

If there exists a perfect permutaion matrix, $\mathbf{P}^*$, then there exists such $\mathbf{S}^*$, s.t.

$$\mathrm{tr}(\mathbf{P}^{*\top}\mathbf{U}_{M_{(s,t)}}\mathbf{S}^*\mathbf{U}_{K_i}^\top) = \mathrm{tr}(\mathbf{P}^{*\top}\mathbf{P}^*) = n \tag{13}$$

Thus, based on Equation (11), Equation (12) and Equation (13), we can get

$$\mathrm{tr}(\mathbf{P}^{*\top}|\mathbf{U}_{M_{(s,t)}}||\mathbf{U}_{K_i}^\top|) \leq n. \tag{14}$$

This means that $\mathbf{P}$ maximizes $\mathrm{tr}(\mathbf{P}^\top|\mathbf{U}_{M_{(s,t)}}||\mathbf{U}_{K_i}^\top|)$ since $\mathrm{tr}(\mathbf{P}^\top|\mathbf{U}_{M_{(s,t)}}||\mathbf{U}_{K_i}^\top|)$ for any permutation matrix P. Therefore, when $\mathbf{K}_i$ and $\mathbf{M}_{(s,t)}$ are isomorphic, the optimal permutation matrix can be obtained as a permutation matrix $\mathbf{P}$ which maximizes $\mathrm{tr}(\mathbf{P}^\top|\mathbf{U}_{M_{(s,t)}}||\mathbf{U}_{K_i}^\top|)$. Therefore, we take $\mathbf{P}^* = |\mathbf{U}_{\mathbf{M}_{(s,t)}}||\mathbf{U}_{K_i}^\top|$ directly.

### 7.2 AN EXAMPLE FOR DEEP ISOMORPHIC NEURAL NETWORK

To better illustrate the idea of our deep model, we also provide the model architecture involves two graph isomorphic feature extraction components. Suppose the kernel size for the first graph isomorphic layer is $k_1$ with $c$ channels, whereas the kernel size of the second graph isomorphic layer is $k_2$ with $m$ channels. The model is shown in Figure 6. After the first graph isomorphic feature extraction component, we get the first feature tensor $\mathcal{Q}_1$ and each element in $\mathcal{Q}_1$ denotes matching

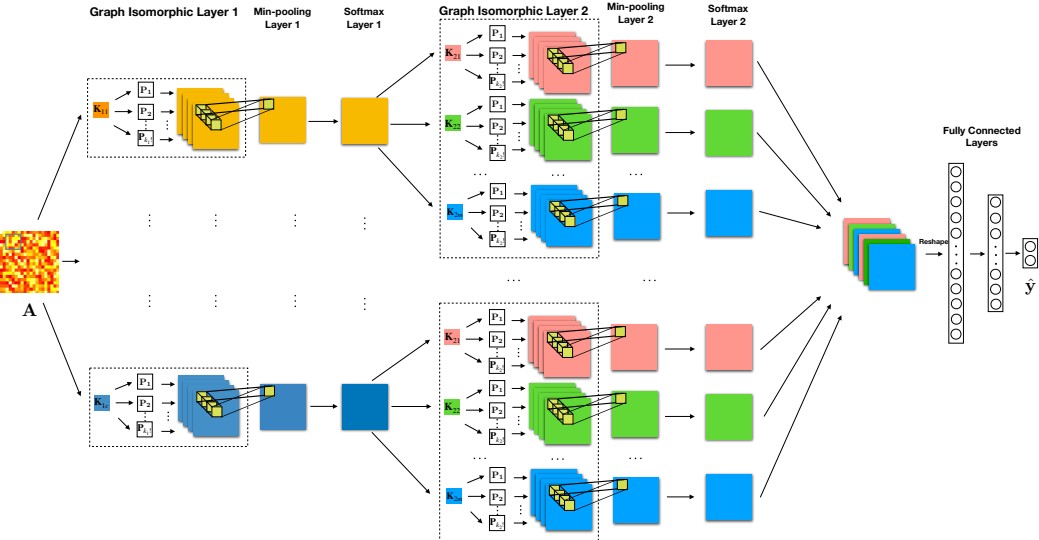

Figure 6: Deep IsoNN Framework Architecture with Two Graph Isomorphic Layers.

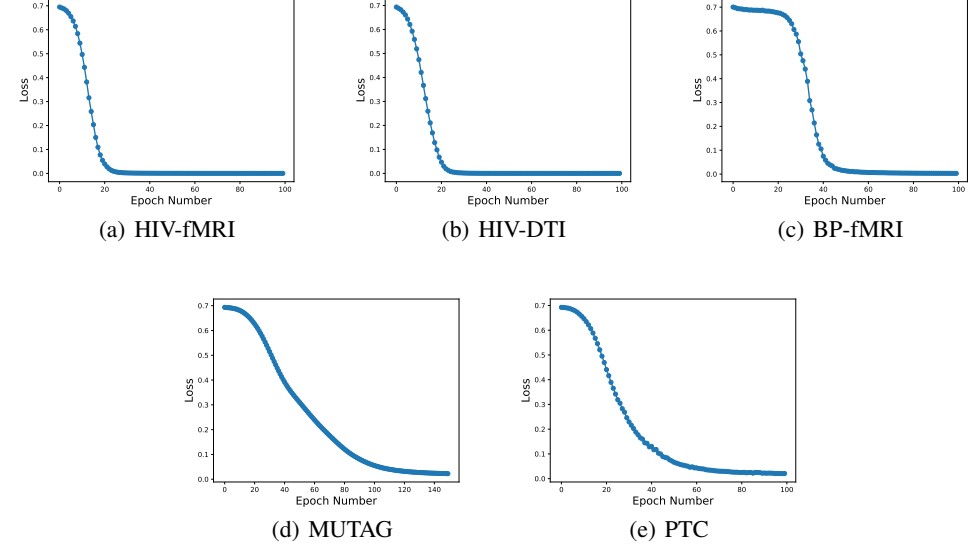

(a) HIV-fMRI  (b) HIV-DTI  (c) BP-fMRI

(d) MUTAG  (e) PTC

Figure 7: Convergence Analysis

score between one subgraph to one kernel template. Thus, we can also regard each element in $\mathcal{Q}_1$ as a kernel template. Since we have $c$ channel in the first component, the second component will be used on every channel of $\mathcal{Q}_1$. If the channel number of the second component is $m$, then the first dimension of the learned feature tensor $\mathcal{Q}_2$ of the second component is $c * m$. For a deeper model with 3 or more graph isomorphic feature extraction components, our will do similar operations to the second isomorphic components. The first dimension of the final tensor $\mathcal{Q}$ will be the product of channels in all former graph isomorphic layers.

### 7.3 CONVERGENCE ANALYSIS

The Figure 7 shows the convergence trend of ISONN on five datasets, where the x-axis denotes the epoch number and the y-axis is the training loss, respectively. From these sub-figures, we can know that the proposed method can achieve a stable optimal solution within 50 iterations except for MUTAG (it needs almost 130 epochs to converge), which also illustrates our method would converge relatively fast.

