# OpenReview forum: "IsoNN: Isomorphic Neural Network for Graph Representation Learning and Classification"
_ICLR.cc/2020/Conference — Reject_

### Official Review · AnonReviewer3 · 2019-10-17
**Official Blind Review #3**

**Rating:** 6

**Review:**

This paper proposes a neural network architecture to classify graph structure. A graph is specified using its adjacency matrix, and the authors prose to extract features by identifying temples, implemented as small kernels on sub matrices of the adjacency matrix. The main problem is how to handle isomorphism: there is no node order in a graph. The authors propose to test against all permutations of the kernel, and choose the permutation with minimal activation. Thus, the network can learn isomorphic features of the graph. This idea is used for binary graph classification on a number of tasks.

Graph classification is an important problem, and I found the proposed solution to be quite elegant. The paper is mostly well written (it could use some proofreading, but the main ideas are explained well). Overall, I liked the idea and tend towards acceptance.

In the experiments, the authors report using different hyper parameters for each data set (e.g., k). I did not understand how these parameters were chosen, since only training and testing sets were reported. I would like the authors to clarify how model selection was performed.

Also, Figure 1 and the details in Section 4 discuss a 1-layer isomorphic NN. The discussion in Section 4.3.2 discusses multi-layer feature extraction. If I understand correctly, this means to apply the graph isomorphic layer + min pooling + softmax several times, but this should be stated explicitly.

**Experience Assessment:**

I do not know much about this area.

**Review Assessment: Checking Correctness Of Derivations And Theory:**

I did not assess the derivations or theory.

**Review Assessment: Checking Correctness Of Experiments:**

I assessed the sensibility of the experiments.

**Review Assessment: Thoroughness In Paper Reading:**

I made a quick assessment of this paper.

---

> ### Author Response · Authors · 2019-11-15
> **Response to Reviewer #3**
>
> We thank the reviewer for the comments and appreciation, and would like to answer the reviewer’s questions as follows:
>
> Q1. In the experiments, the authors report using different hyperparameters for each data set (e.g., k). I did not understand how these parameters were chosen since only training and testing sets were reported. I would like the authors to clarify how the model selection was performed.
>
> Since we have conducted experiments on the relatively small datasets, we try different parameters (i.e., k, c) to train several different models. According to the performances on the testing set of different models with different parameters, we select the model that has the best performance on the testing set. If there is a big dataset, we can split the dataset into training, validation and testing set, choosing the parameters according to the model performance on the validation set.
>
> Q2. Figure 1 and the details in Section 4 discuss a 1-layer isomorphic NN. The discussion in Section 4.3.2 discusses multi-layer feature extraction. If I understand correctly, this means to apply the graph isomorphic layer + min pooling + softmax several times, but this should be stated explicitly.
>
> We are grateful for your advice. Indeed, multi-layer feature extraction means multiple graph isomorphic feature extraction components. We will take the output of the former feature extraction component as the input of the latter feature extraction component. Note that each feature extraction component contains the “graph isomorphic layer + min pooling layer + softmax layer”, which means the deep architecture of the multi-layer feature extraction is “(graph isomorphic layer + min pooling layer + softmax layer) + … + (graph isomorphic layer + min pooling layer + softmax layer)”. Since graph isomorphic layer is the main functional layer to learn subgraph features, we simply use the multi-layer for short in section 4.3.2. We will clarify it in section 4.3.2.  In addition, we also provide an example in the appendix to facilitate your understanding.
>
>
> Hope our response has resolved your concerns. If there is any proposed question about this paper not resolved in our response, welcome to let us know and we are happy to discuss more with you.

---

### Official Review · AnonReviewer1 · 2019-10-23
**Official Blind Review #1**

**Rating:** 3

**Review:**

This paper proposes a method to learn graph features by means of neural networks for graph classification.
In the proposed method, a graph is described by bag of sub-graphs and the sub-graph dictionary is learned through isomorphic matching.
The authors present two approaches toward the isomorphic matching; one is a brute-to-force approach to check all the node permutations and the other is based on spectral decomposition toward efficient computation.
In the experiments on the graph classification tasks using several benchmark datasets, the learned features by the proposed method exhibit favorable performance in comparison with the other graph-based methods.

This paper is leaning toward rejection because (1) the proposed method lacks novelty, (2) it contains technically imprecise parts and (3) the effectiveness is not fully validated in the experiments.
The detailed comments are as follows.

* The presented method belongs to the standard feature representation framework that describes graphs by bag of sub-graph templates (dictionary) [a], and this paper's contribution can be found in the way to learn sub-graph dictionary as in learning convolution kernels of CNNs; in contrast to CNN, the graph representation poses a challenging issue of "isomorphism". It, however, simply employs brute-to-force approach toward the graph isomorphism, lacking novelty. On the other hand, the alternative approach relaxes graph matching into Eq.(8) through spectral decomposition. But, it seriously degrades the characteristics of the permutation matrix P and thus the resulting score z does not exhibit a graph matching measure anymore. So, Eq.(8) lacks theoretical justification and is far away from the sub-graph based representation; I cannot understand what kind of features are actually extracted by Eq.(8).

* Though the authors insist that the method retains the explicit graph structural information, are any constraints imposed on the kernel K for embedding the graph structure into K? Namely, the kernel K is required to exhibit the nature of adjacency matrix of sub-graph. It lacks description and/or discussion about the aspect.

* The node-order information still exists in the classification layer (Sec.4.2) since the FC classifier is directly applied to the (flattened) feature map (tensor) Q in which two axes are defined according to the node order in the graph. This contradicts the authors' claim that the method is invariant to node ordering. For accomplishing node-orderless classification, the global pooling such as GAP should be applied to the final feature map before the classifier layer. In addition, I cannot fully understand how to stack the sub-graph based feature extraction (Sec.4.1) in a "deep" manner? After extracting the sub-graph representation first, the resulting matrix is just a feature map of c channels, not an adjacency matrix which contains the pair-wise relationships between nodes. It is unclear how to construct the deeper model by repeatedly applying the sub-graph template matching.

* The method is built upon the local kernel (K) over the adjacency matrix (A). Although it is invariant against the node order "locally" within the local kernel, the method cannot capture the sub-graph structures beyond the locally ordered nodes in A; "locally" ordered nodes in A which exhibits certain sub-graph can be easily spread "globally" via applying node permutation to A. Thus, the method is only applicable to the limited case that node orders of input graphs are "roughly" canonicalized. This paper completely lacks discussion nor analysis about such a limitation/assumption of the method regarding locality.

* In the experiments, the classifier modules are different across the comparison methods. The proposed method that is a feature extraction from graphs should be fairly compared with the other types of graph feature extraction methods in a consistent pipeline on basis of the identical classifier module. And, as to WL method, the performance of 52.4 on MUTAG in Table 1 is significantly inferior to 80.88 which is reported in [b].

[a] Wale, N., Watson, I.A. and Karypis, G., Comparison of descriptor spaces for chemical compound retrieval and classification, Knowl Inf Syst (2008) 14:3, pp.347-375
[b] Schlichtkrull M., Kipf T.N., Bloem P., van den Berg R., Titov I., Welling M. (2018) Modeling Relational Data with Graph Convolutional Networks. In: Gangemi A. et al. (eds) The Semantic Web. ESWC 2018. Lecture Notes in Computer Science, vol 10843. Springer, Cham


Minor comments:
- Improper citation format. Use \citep and \citet properly according to the context.
- This is related to the kernel methods of graph-kernel and string-kernel. It would be better to mention those related kernel functions for clarifying the contributions.

**Experience Assessment:**

I have read many papers in this area.

**Review Assessment: Checking Correctness Of Derivations And Theory:**

I assessed the sensibility of the derivations and theory.

**Review Assessment: Checking Correctness Of Experiments:**

I assessed the sensibility of the experiments.

**Review Assessment: Thoroughness In Paper Reading:**

I read the paper thoroughly.

---

> ### Author Response · Authors · 2019-11-15
> **Response to Reviewer #1 (To be continued ... )**
>
> Q5. I cannot fully understand how to stack the sub-graph based feature extraction (Sec.4.1) in a "deep" manner? After extracting the sub-graph representation first, the resulting matrix is just a feature map of c channels, not an adjacency matrix which contains the pair-wise relationships between nodes.
>
> Thank you for pointing it out, we will also revise this part. Note that each graph isomorphic feature extraction component contains “graph isomorphic layer + min pooling layer + softmax layer”, we clarify the deep model is the deep architecture of multi-layer feature extraction component, which is equivalent to “(graph isomorphic layer + min pooling layer + softmax layer) + … + (graph isomorphic layer + min pooling layer + softmax layer)”. Let’s say we have 2 graph isomorphic feature extraction components. After the first graph isomorphic feature extraction component, we get the first feature tensor $\mathcal{Q}_1$ and each element in $\mathcal{Q}_1$ denotes matching score between one subgraph to one kernel template. Thus, we can also regard each element in $\mathcal{Q}_1$ as a kernel template. Since we have $c_1$ channel in the first component, the second component will be used on every channel of $\mathcal{Q}_1$. If the channel number of the second component is $c_2$, then the first dimension of the learned feature tensor $\mathcal{Q}_2$ of the second component is $c_1 * c_2$. Similar to the first component, each element of $Q_2$ can represent the kernel templates in the second component. Because $\mathcal{Q}_2$ is derived from $\mathcal{Q}_1$, it is natural to combine the kernels learned by two components and the process can be illustrated in Figure 2.  In addition, we also provide an example in the appendix to facilitate your understanding.
>
>
>
> Q6. The method is built upon the local kernel (K) over the adjacency matrix (A). Although it is invariant against the node order "locally" within the local kernel, the method cannot capture the sub-graph structures beyond the locally ordered nodes in A; "locally" ordered nodes in A which exhibits certain sub-graph can be easily spread "globally" via applying node permutation to A. Thus, the method is only applicable to the limited case that node orders of input graphs are "roughly" canonicalized. This paper completely lacks discussion nor analysis about such a limitation/assumption of the method regarding locality.
>
> In this paper, the node-order is invariant locally. For your mentioned limitation, we are aware that it exists in the model. However, most subgraph-based models like CNN, WL, GIN all based on the adjacency matrix, which is also randomly ordered. Moreover, there are some works considering reordering $\mathbf{A}$. Our model needs to handle such a variance, but we do believe this is not the major work of this paper. However, we will leave it for future work.
>
>
> Q7. In the experiments, the classifier modules are different across the comparison methods.
>
> For baseline models like AE, CNN, Freq, we do set the same classifier module as the proposed model. For SDBN, GCN, GIN, the main reason we keep the original setting is that they already had fined tuned by their authors and reached good results. We just want to make sure that we are comparing with the baseline models that have the best performance.
>
> Q8. As to the WL method, the performance of 52.4 on MUTAG in Table 1 is significantly inferior to 80.88 which is reported in [b].
>
> For WL methods, we have different settings from [b]. To employ the WL kernel methods, the node label should be a known condition. However, our model is proposed for the graphs that do not have node labels. Thus, to have a fair comparison with all baselines, we do not use the node label (i.e., the atomic types of information of chemical compounds). So, to make a fair comparison with WL, we assign each node a unique label instead of the node type. Sorry for the unclear part, we will make changes to the revision.
>
> We also appreciate the reviewer leave us these comments, and we updated the relevant parts already.
>
> Hope our response has resolved your concerns. If there is any proposed question about this paper not resolved in our response, welcome to let us know and we are happy to discuss more with you.
>
> [a] Vishwanathan, S.V.N., Schraudolph, N.N., Kondor, R. and Borgwardt, K.M., 2010. Graph kernels. Journal of Machine Learning Research, 11(Apr), pp.1201-1242.
> [b] Schlichtkrull M., Kipf T.N., Bloem P., van den Berg R., Titov I., Welling M. (2018) Modeling Relational Data with Graph Convolutional Networks. In: Gangemi A. et al. (eds) The Semantic Web. ESWC 2018. Lecture Notes in Computer Science, vol 10843. Springer, Cham

---

> ### Author Response · Authors · 2019-11-15
> **Response to Reviewer #1**
>
> We thank the reviewer for the comments and appreciation, and would like to answer the reviewer’s questions as follows:
>
> Q1. It, however, simply employs brute-to-force approach toward the graph isomorphism, lacking novelty:
>
> The novelty of the proposed model lies in the isomorphic kernel methods instead of simply brute-to-force contract transform. Most existing works focus on the graph kernel with node labels and the kernels methods like WL or the kernels proposed in [a] only computes the similarities between pairwise graphs. Yet, in this paper, we are handling the graph without node labels. Moreover, we can not only compute the similarity between pairwise graphs but also learn subgraph templates. Our approach is simple, but it solves the isomorphism directly. Even though our approach requires high computation when $k$ is big, we find two alternative ways to avoid such a situation, keep the computation cost within an acceptable range.
>
>
> Q2. Eq.(8) lacks theoretical justification and is far away from the sub-graph based representation:
>
> We propose the fast version of IsoNN to deal with the high time cost when $k$ is big (i.e., k>4). We show the theoretical justification of Eq. (8) in the Appendix. We also add more descriptions at the end of section 7.1 to show that Eq. (8) can be an approximation of the optimal permutation matrix. In fact, if we find the optimal permutation matrix $\mathbf{P}^* \in \{0, 1\}^{k \times k}$ directly (i.e., by Hungarian method), it will cost lots of time even though the learned features will be precise. However, if we relax the $\mathbf{P}^*$ to $\mathbf{P}^* \in [0,1]^{k \times k}$, the time cost will decrease rapidly with the degenerated features, and the performance is close to that of the original model (slow version). Thus, if you do not care about what kernel template will be learned, then Eq. (8) can be used. Otherwise, you can learn the precise features by applying multiple graph isomorphic feature extraction components if the kernel size is big.
>
> Q3. Are any constraints imposed on the kernel K for embedding the graph structure into K? Namely, the kernel K is required to exhibit the nature of the adjacency matrix of sub-graphs. It lacks description and/or discussion about the aspect.
>
> The kernel $\mathbf{K}$ is a learned template, containing the most contributing subgraph structure. The kernel template is used to calculate the matching score between the subgraphs and kernel templates, i.e., to see how similar between the subgraphs and the kernel templates. After the computation, we can also locate where the contributing subgraphs are. Since our model is a general model, we don’t impose any constraint on $\mathbf{K}$ for now.
>
> Q4. The node-order information still exists in the classification layer (Sec. 4.2) since the FC classifier is directly applied to the (flattened) feature map (tensor) Q in which two axes are defined according to the node order in the graph. For accomplishing node-orderless classification, the global pooling such as GAP should be applied to the final feature map before the classifier layer.
>
> In this paper, we claim that the proposed model will eliminate the node-order for subgraphs. Let’s say an extreme situation, if the subgraph is the whole graph, the node-order existing in the whole graph can be eliminated by the isomorphic layer. In addition, after the graph isomorphic feature extraction component, the feature tensor $\mathcal{Q}$ only contains the matching scores between subgraphs and kernel templates, i.e., each element denotes the matching score of a subgraph. Thus, the two axes of $\mathcal{Q}$ don’t represent the node-order, they only represent one possible subgraph order. When we flatten the tensor $\mathcal{Q}$, the subgraph order is changed as well since the “flatten” operation will turn three axes (including the channel dimension) into one. Moreover, if any global pooling layer like GAP is applied, it will either degenerate the representation power of kernel templates or lose the precise features of the subgraphs.

---

### Official Review · AnonReviewer2 · 2019-10-23
**Official Blind Review #2**

**Rating:** 1

**Review:**

This paper proposes a new neural network architecture for dealing with graphs dealing with the lack of order of the nodes. The first step called the graph isomorphic layer compute features invariant to the order of nodes by extracting sub-graphs and cosidering all possible permutation of these subgraphs. There is no training involved here as no parameter is learned. Indeed the only learning part is in the so-called classification component which is a (standard) fully connected layer. In my opinion, any classification algorithm could be used on the features extracted from the graphs.
Experiments are then given for the graph classification. I do not understand results of Table 1 as the accuracies reported for MUTAG and PTC in Xu et al with GIN are much higher than the numbers here.

**Experience Assessment:**

I have read many papers in this area.

**Review Assessment: Checking Correctness Of Derivations And Theory:**

I assessed the sensibility of the derivations and theory.

**Review Assessment: Checking Correctness Of Experiments:**

I assessed the sensibility of the experiments.

**Review Assessment: Thoroughness In Paper Reading:**

I read the paper at least twice and used my best judgement in assessing the paper.

---

> ### Author Response · Authors · 2019-11-15
> **Response to Reviewer #2**
>
> We thank the reviewer for the comments and appreciation, and would like to answer the reviewer’s questions as follows:
>
> Q1. There is no training involved here as no parameter is learned.
>
> Actually, there are learnable variables in the graph isomorphic layer, which are the set of kernel templates $\mathbf{K}_i$s. For example, assume we have one isomorphic feature extraction component, our proposed model will do the following steps:
> •	In the graph isomorphic layer, each kernel template $\mathbf{K}_i$ will result in $k!$ feature matrix with $k!$ permutation matrices, where each element in the feature matrices represents the matching score of one subgraph to the corresponding kernel template with one possible permutation matrix.
> •	Passing all $k!$ feature matrices for all kernel templates into the min-pooling layer in order to find the “optimal” features generated by the optimal node permutation for all kernel templates $\mathbf{K}_i$s.
> •	Next, to rescale the “optimal” features, we apply the softmax layer and get the features that related to the kernel variables to further recognize the subgraphs similar to the templates.
> •	Feeding the final features that related to the kernel variables to the classifier, predicting the labels for graphs in the training set.
> •	Calculating the cross-entropy loss based on the predicted labels and ground truth.
> •	Using the gradient descent algorithm and backpropagation to update the parameters in the classifier and the graph isomorphic layer, i.e., kernel variables $\mathbf{K}_i$s.
>
>
> The novelty of the proposed model lies in the isomorphic kernel methods. Most existing works focus on the graph kernel with node labels and the kernels methods like WL or the kernels proposed in [a] only computes the similarities between pairwise graphs. Yet, in this paper, we are handling the graph without node labels. Moreover, we can not only compute the similarity between pairwise graphs but also learn subgraph templates. Our approach is simple, but it solves the isomorphism directly. Even though our approach requires high computation when $k$ is big ($k$>4), we find two alternative ways to avoid such a situation, keep the computation cost within an acceptable range.
>
>
> Q2. Accuracies reported for MUTAG and PTC in Xu et al with GIN are much higher than the numbers here.
>
> In Xu et al with GIN paper, they predict the graph label by utilizing the node label information for MUTAG and PTC according to their source code [a]. However, our model does not need additional node labels. To make a fair comparison, we hide the node label information. We also indicated this in section 5.1.2.
>
>
> Hope our response has resolved your concerns. If there is any proposed question about this paper not resolved in our response, welcome to let us know and we are happy to discuss more with you.
>
> [a] GIN source code: https://github.com/weihua916/powerful-gnns

---

### Author Response · Authors · 2019-10-10
**Equation reference in Section 5.1.1 for IsoNN-fast**

The question mark in section 5.1.1 on page 7 for IsoNN-fast should refer to Equation (8).
Latex fails to generate that reference, and just wanna to clarify.

---

### Decision · Program_Chairs · 2019-12-19

**Decision:**

Reject

**Comment:**

This paper proposes a method to learn graph features by means of neural networks for graph classification.
The reviewers find that the paper needs to improve in terms of novelty and experimental comparisons.